# The Psychometric Performance of the Clinical Learning Environment, Supervision and Nurse Teacher Scale (CLES+T) Among Nursing Students Undertaking Placements in Regional and Rural Australia

**DOI:** 10.3390/nursrep15120429

**Published:** 2025-12-02

**Authors:** Yangama Jokwiro, Qiumian Wang, Jennifer Bassett, Sandra Connor, Melissa Deacon-Crouch, Edward Zimbudzi

**Affiliations:** 1Department of Rural Health Sciences, La Trobe University, Bendigo, VIC 3550, Australia; m.deacon-crouch@latrobe.edu.au; 2Renal Service, Western at Home, Western Health, Melbourne, VIC 3021, Australia; 3Department of Rural Health Sciences, La Trobe University, Shepparton, VIC 3630, Australia; j.bassett@latrobe.edu.au; 4Department of Rural Health Sciences, La Trobe University, Mildura, VIC 3500, Australia; s.connor@latrobe.edu.au; 5School of Nursing and Midwifery, Monash University, Clayton, VIC 3800, Australia; edward.zimbudzi@monash.edu

**Keywords:** nursing students, education, clinical placement, rural and remote nursing, CLES+T, clinical learning environment, nurse supervision, clinical supervision, professional experience placements

## Abstract

**Background**: Clinical Learning Environments (CLEs) are essential to nursing education as a platform for students to develop professional identity; consolidate knowledge with clinical practice; and to gain cognitive, communication, and psychomotor skills. Experience in CLEs significantly impacts nursing students’ satisfaction with education and graduate career preferences. The Clinical Learning Environment, Supervision and Nurse Teacher scale (CLES+T) is widely used to measure the quality of professional experience placements (PEPs), but it has limited evidence of psychometric performance in rural and regional Australian contexts. **Aim**: To assess the psychometric properties of the CLES+T scale in the Australian context of rural and regional undergraduate nursing PEPs. **Methods**: A cross-sectional observational study of a convenience sample of 165 undergraduate nursing students from regional Victoria, Australia, who undertook PEPs between January and June 2020. Participants completed the CLES+T scale post-PEP. Statistical analyses included a test of survey tool reliability using Cronbach’s alpha and exploratory factor analysis to investigate instrument dimensionality and validity. **Results**: The CLES+T scale displayed adequate validity and reliability levels and demonstrated internal consistency similar to previous studies. The most important factor in the CLE was revealed as “pedagogy atmosphere and the content of supervisory relationship” followed by “role of the nurse educator”. **Conclusions**: The CLES+T shows adequate psychometric properties as a valid tool for use with undergraduate nursing students undertaking PEPs in Australian regional, rural, and remote settings.

## 1. Introduction

Contemporary nursing educational programmes are delivered using a blended learning approach that includes both online and classroom-based theoretical education and applied clinical experience, the latter taking place during professional experience placements (PEPs) in a variety of healthcare settings. Healthcare settings, commonly referred to in the literature as Clinical Learning Environments (CLEs), are essential to nursing education. They serve as immersive spaces where nursing students can translate theoretical knowledge into practical application, refine cognitive, communication, and psychomotor skills, and begin developing their professional identity through real-world clinical experiences [1,2]. Given the significant impact CLEs have on nursing students’ satisfaction with education, learning outcomes, and graduate career preferences [3,4,5,6,7], a significant focus has been put on enhancing the quality of CLEs to ensure that nursing students are well-prepared to join the professional workforce as work-ready graduates. Consequently, instruments for measurement of the quality of CLEs have been extensively studied and evaluated to identify areas of CLEs which need improvement, aiming to reduce barriers as well as promote enablers of quality placement experiences [8,9,10].

Several instruments are available for evaluating the clinical learning environment (CLE) for nursing students, including the Clinical Learning Environment Inventory (CLEI), the Clinical Learning Environment and Supervision (CLES) scale, the Clinical Learning Environment, Supervision and Nurse Teacher (CLES+T) scale, and the Clinical Learning Environment Comparison Survey (CLECS) [8,9]. The CLEI primarily assesses the psychosocial aspects of the learning environment, while the CLECS is designed to compare experiences in traditional and simulated clinical settings [8,9,10]. However, many of these tools have notable limitations, such as insufficient attention to the supervisory relationship, the role of the nurse teacher, or the integration of theory and practice. In the past decade, the number of instruments measuring clinical placement quality has increased significantly, yet the CLES+T scale has emerged as the most widely used tool due to its comprehensive coverage of key domains relevant to nursing education and its strong psychometric properties [11,12].

The Clinical Learning Environment, Supervision and Nurse Teacher (CLES+T) scale is an internationally recognised instrument developed to evaluate nursing students’ perceptions of their clinical learning experiences. Grounded in educational and nursing theories, it highlights the dynamic interplay between learning environments, supervisory support, and educational guidance. Originally created in Finland in the early 2000s, the CLES+T scale expanded upon the earlier CLES instrument [11,12] by incorporating the role of the nurse teacher to recognise the link between theory and practice in nursing education. The scale evaluates key domains including the pedagogical atmosphere, leadership style, nursing care practices, supervisory relationships, and the contribution of the nurse teacher [11,12]. The CLES and CLES+T scale have been widely translated and validated in studies across Europe, Asia, the Middle East, and the Americas, demonstrating strong cross-national and cross-linguistic reliability and supporting their use as internationally comparable instruments for evaluating clinical learning environments [9,10].

Although the CLES+T scale has demonstrated validity and reliability in broader educational contexts [10], its application within regional settings remains unexplored. This presents a critical research gap, as regional healthcare environments may differ significantly in terms of clinical exposure, resource availability, and student support structures. Without testing the tool in these contexts, it is unclear whether it accurately captures student satisfaction or reflects the unique characteristics of regional Clinical Learning Environments. Addressing this gap would enhance the generalisability of the CLES+T scale and ensure its relevance for diverse nursing education settings across Australia. Therefore, this study aimed to investigate the reliability and validity of the CLES+T scale in the Australian context of rural and regional undergraduate nursing professional experience placements (PEPs).

## 2. Methods

### 2.1. Design

This cross-sectional observational study used convenience sampling of undergraduate nursing students in regional campuses in Victoria, Australia. The reporting in this study followed the Strengthening the Reporting of Observational Studies in Epidemiology (STROBE) guidelines [13]. Data were collected from a university within the northern region of Victoria, providing a unique perspective on the clinical learning environment in regional and rural areas in Australia. This focus allowed understanding of the unique challenges and strengths present in rural and regional nursing education, such as workforce shortages, multi-role demands on educators, and strong community engagement. This factor can significantly influence students’ clinical learning experiences and perceptions.

### 2.2. Study Tool

Created by Saarikoski et al., the CLES+T scale is based on their previous work on the Clinical Learning Environment and Supervision scale (CLES) [12]. The CLES+T scale has an additional sub-category to assess the role of nurse educators in clinical practice. Since its development and then validation in Finland [11], the CLES+T scale has been translated across various languages, and its psychometric properties have been assessed and validated in Spain [14], Croatia [15], Turkey [1], Slovakia [16], Austria [17], China [18], Morocco [19], and many other countries. The scale is composed of 34 items, which are categorised into five distinct sub-categories: pedagogical atmosphere, consisting of 9 items; the leadership style of the ward manager, which includes 4 items; nursing care on the ward, with 4 items; the content of the supervisory relationship, encompassing 8 items; and the role of the nurse teacher, which is represented by 9 items [11]. Each of these items is assessed using a five-point Likert scale, where 1 represents “fully disagree,” 2 corresponds to “disagree to some extent,” 3 stands for “neither agree nor disagree,” 4 signifies “agree to some extent,” and 5 indicates “fully agree” [11]. After receiving formal permission from the original author of the CLES+T scale, a minor translation of the English version was carried out. The instrument was adapted by the research team to use terms that describe the Australian context until a coherent version was agreed upon amongst the researchers involved in this study. For example, “ward manager” was changed to “Nurse Unit Manager (NUM),” which is the standard term in Australian hospitals.

### 2.3. Study Sample

Participant inclusion criteria were that the students were enrolled in the Bachelor of Nursing at any of the four regional/rural campuses of a university located in Victoria, Australia. Participants were undertaking either a 2- or 3-year Bachelor of Nursing programme designed for students with previous tertiary or other nursing qualifications. Additionally, these students needed to be undertaking their PEP between January and June 2020 at a variety of regional and rural health services in settings classified as Modified Monash Model (MMM) 2–7. The Modified Monash Model is a geographical classification created on the Australian Statistical Geography Standard-Remoteness Areas (ASGS-RS) framework and is used by Australian Government to describe settings as metropolitan (MMM1), rural, remote, or very remote (MMM2–7) [20]. Exclusion criteria were undertaking PEPs at an MMM1, a Metropolitan hospital. Of the 385 eligible students, 170 consented to participate, a response rate of 44%, and 165 (N = 165) met the eligibility criteria to be included in study.

### 2.4. Data Collection

The study questionnaire was uploaded into the Qualtrics Survey platform [21]. On completion of their PEP, students were invited to participate in the study by a clinical placement officer not affiliated with the research. Invitations were sent via email, accompanied by regular reminders to encourage survey completion. An information statement detailing the project and emphasising the voluntary nature of participation was provided to participants; participant consent was indicated through completion of the questionnaire. No personal identification was requested, and access to an anonymous questionnaire was administered through Qualtrics.

### 2.5. Data Analysis

The CLES+T scale [11] was tested for reliability and validity. Reliability was evaluated by Cronbach’s alpha (>0.7), item–total correlations (>0.3), and inter-item correlations (0.2–0.4) [22,23]. The Kaiser–Meyer–Olkin (KMO) measure of sampling adequacy, along with Bartlett’s test of sphericity, were analysed initially to determine suitability of the data for conducting a factor analysis. The KMO test was evaluated with a cut-off range greater than 0.6 and less than 1.0, while Bartlett’s test required statistical significance, specifically with a *p*-value of less than 0.001, to confirm the appropriateness of the data for factor analysis [24,25,26].

Exploratory factor analysis investigated instrument dimensionality and validity. Principal Component Analysis (PCA) was used initially followed by Principal Axis Factoring (PAF) to focus on the underlying latent factors [11,27]. Eigenvalues were used in factor analysis to determine the number of factors to retain, with the criterion being to retain factors with eigenvalues greater than 1, as they explain more variance than a single observed variable [11,27]. Both orthogonal rotation (Varimax) and oblique rotation (Oblimin) were applied. Orthogonal rotation (Varimax) assumes that the factors do not correlate with each other, so it was used to obtain a set of independent factors, while the oblique rotation (Oblimin) allows for any correlations between factors. The data were statistically analysed using the Statistical Package for Social Sciences (SPSS), Version 24.0 (SPSS, Chicago, IL, USA).

### 2.6. Ethical Considerations

Ethics approval was granted by both the University’s Human Research Ethics Committee and the Human Research Ethics Committee of the healthcare agencies involved (HREC19409). The explanatory statement provided to participants outlined the purpose of the study, the voluntary nature of participation, and that they could withdraw from the study without any repercussions for their clinical practice. Additionally, no personally identifying data was required, and confidentiality was strictly maintained. Information on data storage and complaint procedures was also provided to participants [28].

## 3. Results

### 3.1. Sample Characteristics

The questionnaire was completed by 165 eligible and consenting students. Table 1 presents the demographic characteristics of the sample. Most participants (68%) were aged 30 years or younger, with the largest group (43%) in the 21–30 age range; among these students the majority (88%) identified as female. Nearly three-quarters (72%) were enrolled in the undergraduate 3-year Bachelor of Nursing programme, and a substantial proportion (85%) were in their second or third year of study at the time of their placement. Clinical placements were primarily undertaken in public healthcare services (87%), with a smaller proportion in private settings (13%). Most students (75%) completed their placements in regional (MMM 2) or large rural town (MMM 3) locations, while the remainder were placed in MMM 4 (11%), MMM 5 (12%), or MMM 7 (1%) areas. The majority of participants (74%) were supervised by an educator or facilitator during their last placement, while 8% were supervised by Enrolled Nurses or Personal Care Attendants, not meeting the minimum standard for Registered Nurse supervision. Most students (74%) did not have a consistent nurse supervisor, and approximately 19% reported not receiving any scheduled private supervision meetings.

### 3.2. Item Performance

Table 2 shows the item performance of the CLES+T scale. The distribution of items was negatively skewed with relatively few values on the lower end of the scale and a leptokurtic distribution. Out of the 34 items of the CLES+T scale [11,12], 31 met the cut-off criteria for item–total correlations above 0.3 as shown in Table 2. The three items not meeting the cut-off criteria and therefore excluded were (i) the common meetings between myself, buddy nurse, and nurse educator were a comfortable experience, (ii) in our common meetings I felt that we are colleagues, (iii) focus on the meetings was on my learning needs. However, the three item outliers with much lower means and item–total correlations suggest a specific area of the clinical learning environment that may need targeted improvement or further qualitative exploration. Cronbach’s alpha of 0.94 and the Omega of 0.92 showed satisfactory internal consistency reliability. Cronbach’s alpha increased by deleting the three items that did not meet the item–total correlation criteria mentioned above. Reliability (Cronbach’s alpha) of the subscales after factor analysis were Factor 1 (0.95), Factor 2 (0.94), Factor 3 (0.84), Factor 4 (0.88), Factor 5 (0.73), and Factor 6 (0.62), indicating that Factor 6 did not achieve an adequate reliability score to be included [22,23].

### 3.3. Exploratory Factor Analysis

Three items related to common meetings between students, buddy nurses, and nurse educators did not meet the cut-off criteria for corrected item–total correlation (Table 2) and were excluded from the exploratory factor analysis (EFA) to improve the reliability of the scale. Sample adequacy for the remaining 31 items was confirmed by a Kaiser–Meyer–Olkin (KMO) test value of 0.92 and a highly significant Bartlett’s test of sphericity (χ^2^ = 4208.48, df = 465, *p* < 0.001), indicating substantial correlation among items and suitability for factor analysis.

Principal Component Analysis (PCA) followed by Principal Axis Factoring (PAF) were conducted to identify the underlying latent factors, consistent with best practice for scale validation in nursing education [11,27]. PAF was chosen as it does not rely on distributional assumptions, which is appropriate given the exclusion of three items with poor performance. The Kaiser criterion (eigenvalue > 1) was applied [22,29], alongside Cattell’s scree test and consideration of the percentage of variance explained, with each factor required to account for at least 5% of the variance.

PAF on the 31 items revealed six latent factors, collectively explaining 67% of the total variance (Table 3). PCA also retained six factors, accounting for 73% of the variance, demonstrating stability in the factor structure. Both orthogonal (Varimax) and oblique (Oblimin) rotations were applied; however, Varimax results are reported to present independent factors. The factor loadings showed some cross-loading, particularly for Factor 6, which contained only two items that also loaded onto Factor 1.

The most significant factor influencing the clinical learning environment was “pedagogy atmosphere and the content of supervisory relationship,” with an original eigenvalue of 13.6 (6.85 after rotation) and explaining 22% of the variance post-rotation. The next most prominent factor was the “role of nurse educator,” with an eigenvalue of 4.63 and accounting for 15% of the variance after rotation. Most items loaded adequately onto Factor 1, reflecting the centrality of pedagogical atmosphere and supervisory relationships in student learning experiences (Table 3).

It is important to note that Factor 6 did not achieve an adequate reliability score (Cronbach’s alpha = 0.62) [22,23], and while it is included in Table 3 for transparency, it was not interpreted further in the results.

## 4. Discussion

This study examined the psychometric properties of the Clinical Learning Environment, Supervision and Nurse Teacher (CLES+T) scale [11,12] among Australian nursing students undertaking placements in rural and regional settings. The CLES+T scale demonstrated strong validity and reliability, with an overall Cronbach’s alpha of 0.94, consistent with previous validation studies across diverse international contexts [14,27,30,31,32]. This high internal consistency supports the scale’s robustness for evaluating clinical learning environments in regional and rural Australia.

A key finding in this study is the high prevalence of inconsistent supervision in rural and regional placements. This study reports that 74% of students did not have a consistent personal supervisor during their placement, which is higher than that reported in many metropolitan or international studies [27,33]. This inconsistency is likely a reflection of workforce shortages and the multi-role demands placed on nurse educators in rural settings [6]. Furthermore, nearly one in five students (19%) reported not receiving any scheduled private supervision meetings, highlighting a significant gap in structured support for students in these contexts.

Another unique aspect observed was the pronounced ceiling effect in item responses (Table 2), with most item means ranging from 4.01 to 4.62, and negative skewness and leptokurtosis indicating clustering of high scores. While ceiling effects have been noted in other studies [14,27,30], the extent observed here may be influenced by the strong community ties and social cohesion characteristic of rural and regional Australia, potentially leading to more socially desirable responses. This phenomenon may mask underlying challenges or dissatisfaction and suggests that the CLES+T scale may have reduced sensitivity to detect subtle changes or intervention effects in these settings. This should be considered when researching intervention that improves the experience of students in these settings.

Additionally, items relating to common meetings between students, buddy nurses, and nurse educators had the lowest mean scores (2.78–2.99, Table 2), indicating that such collaborative learning opportunities are infrequent or ineffective in rural/regional settings. These meetings are often highlighted as best practice in the clinical education literature [6] yet are not being realised in these contexts. The reliability analysis further identified these items as not meeting the item–total correlation threshold, and their exclusion increased overall reliability to 0.95.

However, students in rural or regional placements highly valued the role of the nurse educator/facilitator in bridging the theory-practice gap (Table 3). This finding reinforces the critical importance of investing in nursing educator roles in non-metropolitan areas, where their impact may be even more pronounced due to the lack of other support.

Factor analysis yielded a six-factor structure, explaining 67% of the total variance (Table 3), which is comparable to the variance explained in previous studies (64–73%) [11,14,30,32]. The most prominent factor was the “supervisory relationship,” followed by the “role of the nurse educator,” both of which showed strong loadings and are consistent with the theoretical underpinnings of the CLES+T scale [11,14,27,31]. The importance of individualised supervision is further supported by evidence that students are more satisfied with one-to-one supervision compared to group models [27,33]. The high endorsement of the nurse educator’s role in this study reinforces its critical function in bridging the theory-practice gap for students in rural placements. Although six factors were identified, only five met the threshold for adequate reliability (Cronbach’s alpha ≥ 0.70). Factor 6, with an alpha of 0.62, was included in Table 3 for transparency but was not interpreted further, in line with best practice in psychometric reporting [22,23]. The variability in factor structures reported in the literature, which range from four to seven factors [7,11,14,31,32], suggests that the dimensionality of the CLES+T scale may be context dependent. Nevertheless, the five-factor solution observed here closely aligns with the theoretical model proposed by Saarikoski et al. [11]. This study found that the five-factor solution derived from exploratory factor analysis closely matches the theoretical model of the CLES+T scale proposed by Saarikoski et al. [11]. The identified factors that are pedagogical atmosphere, supervisory relationship, role of the nurse educator, leadership in the learning environment, and learning opportunities reflect the core domains of the original model and demonstrated strong reliability and item loadings (Table 3). This alignment confirms that the CLES+T scale effectively captures the essential elements of the clinical learning environment for nursing students in rural and regional Australia. The results reinforce the instrument’s validity and suitability for evaluating clinical placements in diverse educational and healthcare contexts. While a sixth factor emerged, it did not meet reliability standards and was not interpreted further, supporting the robustness of the five-factor structure in this setting.

Therefore, even though the factor structure remains far from being settled, there is no adequate evidence to refute the theoretical structure provided by Saarikoski et al. [11] in comparative studies. In this study, the CLES+T scale showed robustness and adequate validity and reliability evaluating the perceptions of student nurses completing PEPs in regional, rural, and remote settings. Use of this tool will allow systematic processing of empirical data collected using valid and reliable instruments to improve resources for student nurses studying in regional, remote, and rural settings. To further validate the factor structure identified in this study, future research with a larger sample or multicentre study could employ Confirmatory Factor Analysis (CFA) [34].

### 4.1. Limitations and Strengths

The convenience sampling method utilised in this study represents a significant limitation. Specifically, student recruitment was conducted from only one region of Australia and a single educational institute. Future research should aim to expand recruitment across multiple regions and diverse educational settings to better capture the variability in student experiences. Incorporating stratified or random sampling methods would further enhance representativeness and reduce selection bias. Additionally, multi-site collaboration and comparative analysis across institutions could provide richer insights and strengthen the applicability of results to national education and health contexts. The cross-sectional nature combined with the specific contextuality of the data requires careful interpretation, as the findings may not be directly translatable to other settings and contexts [35]. Furthermore, the reliance on self-reporting data may potentially be subject to recall and selection bias [35], and the relative criteria used for reliability and dimensionality is far from being absolute [34]. Therefore, an analysis of further data from additional contexts and regions is essential.

Despite these limitations, there are numerous strengths inherent in this study. Notably, student participation in the online survey was 44% higher than initially anticipated, which is consistent with findings from other studies of this nature [3,4,5,7]. Additionally, the distribution of students who participated is normalised across both year levels at the university and levels of experience. Importantly, there is also a normalised spread of students across the Modified Monash Mode 2 to 7 rurality classifications.

### 4.2. Clinical Implications

This study validated the CLES+T scale for undergraduate nursing students in regional and remote Australia where robust evaluation tools have been notably absent. The analysis reveals that CLES+T not only demonstrates strong reliability and a clear six-factor structure, but also captures the complexities of rural clinical education, with domains like supervision, pedagogical atmosphere, and the nurse educator’s role standing out for their consistency and impact. The findings empower educators and health services to use CLES+T for targeted quality improvement, benchmarking, and workforce development in rural settings. This insight from this work is that while the scale’s core strengths translate well to the rural context, certain items related to collaborative meetings highlight unique challenges and opportunities for adaptation. This highlights the need for ongoing, context-specific evaluation of international tools to ensure that they remain both relevant and applicable as rural clinical education continues to evolve with technology.

## 5. Conclusions

The CLES+T scale demonstrated adequate psychometric properties in this sample of nursing students completing placements in regional, rural, and remote Australia. Its use enables systematic evaluation of clinical learning environments, providing valuable data to inform improvements in supervision, resource allocation, and support for nursing students in non-metropolitan settings. The novel findings regarding supervision continuity, the ceiling effect, and the critical role of nurse educators highlight the unique challenges and opportunities present in rural and regional clinical education.

## Figures and Tables

**Table 1 nursrep-15-00429-t001:** Demographic characteristics.

Demographic Variables	Number (%)
**Age (years)**	
10–20	42 (25.25)
21–30	71 (43.03)
31–40	33 (20.00)
41–50	16 (9.70)
Over 50	3 (1.82)
**Gender**	
Male	19 (11.51)
Female	145 (87.88)
Prefer not to say	1 (0.60)
**Course**	
Undergraduate 3-year BN programme	118 (71.51)
Graduate entry 2-year BN programme	19 (11.52)
Enrolled Nurse Conversion BN programme	28 (16.97)
**Year level on completion of last placement**	
First Year	24 (15.15)
Second Year	82 (49.70)
Third Year	59 (35.76)
**Modified Monash Model Classification of Placement location**	
MMM 2	54 (32.73)
MMM 3	71 (43.03)
MMM 4	18 (10.91)
MMM 5	20 (12.12)
MMM 6	-
MMM 7	2 (1.21)
**Hospital status**	
Public	143 (86.67)
Private	22 (13.33)
**Occupation of supervisor**	
Personal Care Attendant/AIN	3 (1.82)
Enrolled Nurse	11 (6.70)
Registered Nurse	2 (1.21)
Educator/Facilitator	122 (74.00)
Associate Nurse Manager	2 (1.21)
Nurse Manager	25 (15.15)
**Occurrence of supervision**	
Did not have supervisor at all	4 (2.42)
A personal supervisor was named, but the relationship did not work	9 (5.45)
Named supervisor changed during the placement	4 (2.42)
Supervisor varied according to shift or place of work	122 (74.00)
Same supervisor had several students and was a group supervisor	2 (1.21)
Personal supervisor was named and our relationship worked	24 (14.55)
**Frequency of separate private unscheduled supervision**	
Not at all	31 (18.79)
Once or twice during the course	42 (19.39)
Less than once a week	12 (7.27)
About once a week	29 (19.58)
More often	51 (30.91)

**Table 2 nursrep-15-00429-t002:** Item performance of the Clinical Learning Environment, Supervision and Nurse Teacher scale (CLES+T) among nursing student training in regional and remote places in Victoria.

Item	Mean	SD	Skewness	Kurtosis	Corrected Item–Total Correlation	Cronbach’s Alpha If Item Deleted
The staff were easy to approach	4.33	0.89	−1.67	2.84	0.57	0.939
I felt comfortable going to the ward at the start of my shift	4.28	0.97	−1.44	1.44	0.69	0.937
During staff meetings (e.g., before shifts) I felt comfortable taking part in the discussions	3.78	1.09	−0.77	−0.13	0.62	0.938
There was a positive atmosphere on the ward	4.19	1.00	−1.36	1.46	0.78	0.937
The staff were generally interested in student supervision	4.01	1.12	−1.05	0.24	0.65	0.938
The staff learned to know the student by their personal names	4.37	0.86	−1.48	1.92	0.56	0.939
There were sufficient meaningful learning situations on the ward	4.32	0.90	−1.71	3.40	0.59	0.938
The learning situations were multi-dimensional in terms of content	4.18	0.99	−1.37	1.53	0.60	0.938
The ward can be regarded as a good learning environment	4.41	0.90	−1.90	3.87	0.66	0.938
The nurse unit manager (NUM) regarded the staff on her/his ward as a key resource	4.41	0.81	−1.59	3.00	0.54	0.939
The NUM was a team member	4.40	0.83	−1.26	1.12	0.57	0.939
Feedback from the NUM could easily be considered as a learning situation	4.02	1.05	−0.73	−0.22	0.55	0.939
The effort of individual employees was appreciated	4.49	0.80	−1.65	2.59	0.59	0.939
The ward’s nursing philosophy was clearly defined	4.15	0.91	−0.96	0.42	0.45	0.940
Patients received individual nursing care	4.62	0.60	−1.67	3.27	0.50	0.939
There were no problems in the information flow related to patients’ care	4.28	0.92	−1.46	1.92	0.60	0.938
Documentation of nursing (e.g., nursing plans, daily recording of nursing procedures, etc.) was clear	4.48	0.83	−1.93	4.05	0.41	0.940
My supervisor showed a positive attitude towards supervision	4.31	0.88	−1.46	2.13	0.61	0.938
I felt that I received individual supervision	4.38	0.80	−1.46	2.84	0.59	0.939
I continuously received feedback from my supervisor	4.26	0.99	−1.52	1.98	0.71	0.937
Overall, I am satisfied with the supervision I received	4.36	0.96	−1.81	3.26	0.81	0.936
The supervision was based on a relationship of equality and promoted my learning	4.32	0.88	−1.61	2.95	0.72	0.937
There was a mutual interaction in the supervisory relationship	4.46	0.72	−1.64	3.88	0.69	0.938
Mutual respect and approval prevailed in the supervisory relationship	4.38	0.88	−1.81	3.80	0.77	0.937
The supervisory relationship was characterised by a sense of trust	4.40	0.81	−1.70	4.08	0.72	0.938
In my opinion, the nurse educator/facilitator was capable of integrating theoretical knowledge and everyday practice of nursing	4.48	0.86	−2.01	4.07	0.54	0.939
The nurse educator/facilitator was capable of operationalising the learning goals of this clinical placement	4.45	0.91	−2.04	4.06	0.55	0.939
The nurse educator/facilitator helped me to reduce the theory-practice gap	4.42	0.92	−1.81	3.11	0.60	0.938
The nurse educator/facilitator was like a member of the nursing team	4.06	1.17	−1.19	0.47	0.65	0.938
The nurse educator/facilitator was able to give his or her pedagogical expertise to the clinical team	4.15	1.10	−1.27	0.97	0.61	0.938
The nurse teacher and the clinical team worked together in supporting my learning	4.18	1.04	−1.31	1.18	0.71	0.937
The common meetings between myself, buddy nurse and nurse educator were a comfortable experience	2.99	1.54	−0.03	−1.43	0.15	0.945
In our common meetings I felt that we are colleagues	2.78	1.41	0.12	−1.22	0.15	0.945
Focus on the meetings was in my learning needs	2.90	1.44	0.02	−1.29	0.10	0.945

SD = Standard deviation.

**Table 3 nursrep-15-00429-t003:** Factor solution of the Clinical Learning Environment, Supervision and Nurse Teacher scale (CLES+T) among nursing student training in regional and remote places in Victoria.

Item	Factor 1 (Pedagogy Atmosphere and Supervision Content)	Factor 2 (Role of Nurse Educator)	Factor 3 (Leadership in Learning Environment)	Factor 4 (Learning Opportunities)	Factor 5 (Patient Caring Valued)	Factor 6
The staff were easy to approach	0.377				0.344	0.359
I felt comfortable going to the ward at the start of my shift	0.480				0.318	
During staff meetings (e.g., before shifts) I felt comfortable taking part in the discussions	0.370					0.520
There was a positive atmosphere on the ward	0.434			0.484	0.322	0.355
The staff were generally interested in student supervision	0.504			0.441		0.484
The staff learned to know the student by their personal names	0.334					0.421
There were sufficient meaningful learning situations on the ward				0.735		
The learning situations were multi-dimensional in terms of content				0.693		
The ward can be regarded as a good learning environment	0.344	0.304		0.652		
The nurse unit manager (NUM) regarded the staff on her/his ward as a key resource			0.806		0.308	
The NUM was a team member			0.830			
Feedback from the NUM could easily be considered as a learning situation			0.639			
The effort of individual employees was appreciated	0.397		0.323			
The ward’s nursing philosophy was clearly defined			0.454		0.337	
Patients received individual nursing care	0.313				0.519	
There were no problems in the information flow related to patients’ care	0.367				0.706	
Documentation of nursing (e.g., nursing plans, daily recording of nursing procedures, etc.) was clear					0.527	
My supervisor showed a positive attitude towards supervision	0.660					
I felt that I received individual supervision	0.640					
I continuously received feedback from my supervisor	0.779					
Overall, I am satisfied with the supervision I received	0.778					
The supervision was based on a relationship of equality and promoted my learning	0.861					
There was a mutual interaction in the supervisory relationship	0.827					
Mutual respect and approval prevailed in the supervisory relationship	0.795					
The supervisory relationship was characterised by a sense of trust	0.788					
In my opinion, the nurse educator/facilitator was capable of integrating theoretical knowledge and everyday practice of nursing		0.817				
The nurse educator/facilitator was capable of operationalising the learning goals of this clinical placement		0.823				
The nurse educator/facilitator helped me to reduce the theory-practice gap		0.836				
The nurse educator/facilitator was like a member of the nursing team		0.781				
The nurse educator/facilitator was able to give his or her pedagogical expertise to the clinical team		0.777				
The nurse teacher and the clinical team worked together in supporting my learning	0.331	0.735				
Eigenvalues (after rotation)	6.852	4.632	2.753	2.716	2.131	1.577
% of variance explained (total = 67.405%) (after rotation)	22.104	14.943	8.881	8.761	6.873	5.087
Reliability (Cronbach’s alpha)	0.95	0.94	0.84	0.88	0.73	0.62

## Data Availability

The data sets used to support the conclusions of this article are included as tables in the Section 3.

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
