# Peer review of "The Psychometric Performance of the Clinical Learning Environment, Supervision and Nurse Teacher Scale (CLES+T) Among Nursing Students Undertaking Placements in Regional and Rural Australia"

_nursrep, 2025, doi:10.3390/nursrep15120429_

Round 1
Reviewer 1 Report
Comments and Suggestions for Authors
Refer to attached review report
Consider presentation of results in graphs

Reviewer 2 Report
Comments and Suggestions for Authors
Thank you very much for the opportunity to review your manuscript that investigates such an interesting topic. However, I have several concerns and comments that I suggest should be addressed.
General comments
The manuscript does not meet the guidelines and requirements of the Nursing Report Journal. In text-citations and references should be corrected. Guidelines require numbered citations in square brackets and references should be numbered in a list in order of appearance.
Also back matter is missing: Acknowledgments, Author Contributions, Conflicts of Interest
Comments
Abstract
I suggest including number of participants, the time period the research took place and the place in the methods section
In the results section I suggest to include key indices i.e percentage of variance , number of retained factors etc.
Check also for formatting.
I suggest to relocate the “Summary of relevance” to a new more expanded form into a clinical implication section after the discussion. The paragraphs titled “problem or Issue”, “What is already known” and “what this paper adds” should be integrated in the introduction/background section. In their present form, it is a repetition of the study aim.
Introduction
The reason why it is important to use CLES+T in nursing education is limited. Needs to be described explicitly.
Also I suggest expanding the impact of CLE on nursing students’ satisfaction, preferences etc.
The theory of how the CLES tool was developed should also be described.
The introduction section does not contextualize the CLES+T tool against other comparable instruments. It is needed to add an overview of the existing tools and specify why you chose this tool. Did the other tools have any limitations that made you choose this in specific?
Although you mention the use of this tool in different countries, in the Introduction section you under-cite cross-national, cross-linguistic use of CLES/CLES+T. I suggest to add a brief paragraph summarizing translations and validation findings across countries.
Even though you state a gap (the existence of the CLES+T in Australian placements) the why this validation is important is lightly argued. You repeat the gap statement instead of presenting specific stakes for measurement.
Methods
In the study tool sub-section you mention that you did a minor translation which is unclear. You should report on the cross-cultural adaptation process and what was done to ensure content equivalence.
What was the response rate? What was the sample size rationale for EFA, since the 165 respondents are borderline acceptable to run the EFA.
Why did you run both PCA and EFA? Since your aim was to investigate the psychometric properties of the CLES+T, the validation of reflective scale needs a common-factor model like EFA or PAF and not a PCA that adds noise rather than evidence. I suggest to use EFA and then confirm with a CFA and test invariance.
Figure 1 adds little to the study sample. I suggest to omit it.
Ethics: Please provide reference number correctly
Results
Correct descriptive inconsistencies. Check for inconsistencies between text and table. For example, in text you mention public (87%) / private (22%)” vs Table 1 “Public 86.67% / Private 13.33%”. Also, tables should be written according to journal guidelines.
In line 193 you mention that Factor 6 did not achieve adequate reliability score to be included. However, in table 3 includes Factor 6. Please clarify.
Line 219 it is pedagogical atmosphere and not pedagogy. Please check tables too.
Known groups validity was not examined. Please test CLES+T for construct validity and then conduct analysis to control confounding.
You should also report factor correlation to test whether there is a need to merge factors
Limitations
Line 307 Cite these studies
Author Response
See attched

Reviewer 3 Report
Comments and Suggestions for Authors
Reviewer Report for Manuscript ID nursrep-3832370
Comments and Suggestions
- Summary
This manuscript investigates the psychometric properties of the Clinical Learning Environment, Supervision, and Nurse Teacher (CLES+T) scale among undergraduate nursing students undertaking professional experience placements (PEP) in rural and regional Australia. Using a cross-sectional survey of 165 participants, the study employed reliability testing (Cronbach’s alpha) and exploratory factor analysis (EFA). The results indicated that the CLES+T demonstrated acceptable reliability and validity within this specific context, with the pedagogical atmosphere and supervisory relationship emerging as the strongest factors. The study contributes to addressing a notable gap regarding the applicability of the CLES+T scale in rural and regional educational settings.
- General Comments
The paper addresses an important and underexplored area in nursing education, namely the assessment of clinical learning environments in rural and regional settings. The manuscript is clearly structured, adheres to appropriate reporting standards (STROBE), and situates the study within international scholarship. The use of robust statistical methods strengthens the findings. However, the paper would benefit from deeper engagement with theoretical foundations, a more critical discussion of the results, and a clearer articulation of implications for practice and policy.
- Specific Comments
(1) Theoretical Framework: While the introduction highlights the importance of clinical learning environments (CLE) and the widespread use of the CLES+T scale, the theoretical foundation could be expanded to provide stronger grounding. Specifically, incorporating educational and psychological theories—such as experiential learning, social constructivism, or professional identity formation—would help clarify how students acquire knowledge and skills within practice environments. Furthermore, the manuscript should articulate not only how psychometric evaluation establishes the reliability and validity of the CLES+T scale but also how it contributes to broader nursing education theory by informing curriculum design, improving supervision models, and ensuring that educational strategies in clinical settings are evidence-based and theoretically aligned.
(2) Literature Review: The literature review is comprehensive in summarizing validation studies of the CLES+T scale across different countries; however, it would be strengthened by a more critical synthesis that examines how contextual differences—such as urban versus rural settings and cultural variations—may influence the tool’s psychometric performance. While the gap statement identifying the lack of evidence in rural and regional Australia is clear, the authors could further emphasize why these contexts, in particular, necessitate revalidation by highlighting unique challenges such as limited supervision resources, broader role expectations of educators, and differing student-community relationships that may impact the validity and applicability of the scale.
(3) Methodology: Overall, the study demonstrates a sound methodological approach, with the use of exploratory factor analysis (EFA) well justified and the assessment of reliability and validity appropriately supported by relevant references.
(4) Findings: The statistical results are clearly presented and well supported by tables, with the identification of six factors that were subsequently refined into a five-factor solution appropriately justified. However, the manuscript would benefit from a more detailed elaboration on the practical meaning of each factor, particularly by linking them more explicitly to the lived experiences of nursing students in rural placements. Such elaboration would enhance the interpretability of the findings and strengthen their relevance to both educational practice and policy in rural healthcare education.
(5) Discussion and Conclusion: The discussion effectively situates the findings within the context of international studies, which is a notable strength; however, it tends to remain largely descriptive. A more in-depth analysis of why certain items—such as common meetings—performed poorly, particularly in rural contexts, would provide valuable insights and enhance the scholarly contribution. While the conclusion appropriately highlights the potential for national and international comparative studies, it could be further strengthened by offering concrete, practical recommendations for nursing education administrators and policymakers in rural Australia, thereby increasing the study’s relevance and impact on educational practice and policy development.
- Questions and Suggestions
(1) If feasible, the authors could consider discussing how their findings might inform strategies to improve supervision models in rural contexts, where limited resources and educators’ multiple role responsibilities may constrain the overall quality of supervision. Adding such a discussion would strengthen the practical relevance of the study and provide valuable insights for addressing challenges unique to rural nursing education.
(2) If feasible, the paper would benefit from an additional section explicitly outlining the implications of the findings for curriculum design and policy in Australian nursing education. Such a discussion could highlight how validated psychometric tools like the CLES+T can inform the development of evidence-based curricula, strengthen supervision models, and guide policy decisions aimed at improving the quality of clinical learning environments in rural and regional contexts. This would not only enhance the practical significance of the study but also ensure that its contributions extend beyond measurement validation to shaping future educational strategies and workforce planning.
(3) Reviewer’s Comment (for Section 2.6 Ethical Statement):
The Ethical Statement currently reads: “Ethics approval was sought and approved granted by both the University’s Human Research Ethics Committee (refence number, xxxx).”
There are two issues here:
- The word “refence” should be corrected to “reference.”
- The actual ethics approval reference number appears to be missing. If available, please provide the specific approval number issued by the University’s Human Research Ethics Committee.
Kindly revise this section accordingly.
- Overall Evaluation
This manuscript makes a valuable contribution to nursing education research by validating the CLES+T scale in rural and regional Australia, thereby addressing a notable gap in the literature. The methodology is rigorous, and the results align with international findings, reinforcing the robustness of the scale. If feasible, I recommend revisions to deepen the critical discussion of the findings and to articulate clearer practical implications—specifically in line with the suggestions outlined in Section 4, Questions and Suggestions, points (1) and (2). Incorporating these enhancements would further strengthen the contribution of the paper. That said, in its current form, the manuscript is already highly suitable for publication.
Author Response
see atttached

Round 2
Reviewer 2 Report
Comments and Suggestions for Authors
Dear authors,
Thank you for amending and addressing the majority of my comments. However, I would like to state a few more points that need to be addressed.
In the Study sample section you mention that the participants consented to participate were 170 and in the results section you mention 165. Which number is correct?
Also the response rate was very low. This should be reported in the limitations section.
in the study sample you mention MM1 in the exclusion criteria. What does it stand for? probably MMM is the correct acronym? Later in in the strenghts section you mention that as one of your study's strentgh. Please clarify
Factor naming is incosistence across sections: in the abstract you mention “educator relationship" and in the results section you mention "“role of nurse educator"
You state that you run both PCA and PAF and applied both varimax and oblique rotation but only Varimax results are included. Report Oblimin loadings abd the inter-factor correlation matrix.
In the text you mention Omeda of 0.92 it is a typo for Omega
Malformed DOIS several entries contain dublicated prefixes.
In Discussion section there is a yellow highlighted insertion of web link.
Comments on the Quality of English LanguageNeed lanugage checking
